# Pre-Emptive Antimicrobial Locks Decrease Long-Term Catheter-Related Bloodstream Infections in Hemodialysis Patients

**DOI:** 10.3390/antibiotics11121692

**Published:** 2022-11-24

**Authors:** Andres Blanco-Di Matteo, Nuria Garcia-Fernandez, Aitziber Aguinaga Pérez, Francisco Carmona-Torre, Amaya C. Oteiza, Jose Leiva, Jose Luis Del Pozo

**Affiliations:** 1Infectious Diseases Division, Clinica Universidad de Navarra, 31008 Pamplona, Spain; 2Nephrology Department, Clinica Universidad de Navarra, 31008 Pamplona, Spain; 3Clinical Microbiology Department, Hospital Universitario de Navarra, 31008 Pamplona, Spain; 4Navarra Health Research Institute-IdiSNA, 31008 Pamplona, Spain; 5Clinical Microbiology Department, Clinica Universidad de Navarra, 31008 Pamplona, Spain

**Keywords:** bacteremia, chronic kidney failure, catheter-related infection, biofilm, staphylococcus

## Abstract

This study aimed to prove that pre-emptive antimicrobial locks in patients at risk of bacteremia decrease infection. We performed a non-randomized prospective pilot study of hemodialysis patients with tunneled central venous catheters. We drew quantitative blood cultures monthly to detect colonization. Patients with a critical catheter colonization by coagulase-negative staphylococci (defined as counts of 100–999 CFU/mL) were at high risk of developing a catheter-related bloodstream infection. We recommended antimicrobial lock for this set of patients. The nephrologist in charge of the patient decided whether to follow the recommendation or not (i.e., standard of care). We compared bloodstream infection rates between patients treated with antimicrobial lock therapy versus patients treated with the standard of care (i.e., heparin). We enrolled 149 patients and diagnosed 86 episodes of critical catheter colonization by coagulase-negative staphylococci. Patients treated with antimicrobial lock had a relative risk of bloodstream infection of 0.19 when compared with heparin lock (CI 95%, 0.11–0.33, *p* < 0.001) within three months of treatment. We avoided one catheter-related bloodstream infection for every ten catheter-critical colonizations treated with antimicrobial lock [number needed to treat 10, 95% CI, 5.26–100, *p* = 0.046]. In conclusion, pre-emptive antimicrobial locks decrease bloodstream infection rates in hemodialysis patients with critical catheter colonization.

## 1. Introduction

Vascular access is a mainstay in performing efficient hemodialysis (HD) treatment. The maintenance of adequate vascular access and the prevention and treatment of its complications is a challenge. Although arteriovenous fistula remains the first choice for hemodialysis, the long-term tunneled central venous catheter (TCVC) has become the vascular access for many HD patients. According to the United States Renal Data System, 80% of patients begin hemodialysis through a central venous catheter, with 17.6% of prevalent dialysis patients using a central venous catheter in 2018 [1]. The Catalan Renal Registry reported that 32.2% of vascular accesses in 2018 were TCVC [2]. Catheter-related bloodstream infection (CRBSI) increases mortality and hospital cost and impairs patients’ quality of life [3]. CRBSI incidence varies widely across different settings and definitions. In the United States, the rate of CRBSI per 1000 catheter days ranges from 4.69 to 5.55 [4,5]. The Spanish Nephrology Society reported rates between 2.5 and 5 cases/1000 catheters-days in 2017 [6].

Guidelines for the management of CRBSI make no recommendation about the role of routine surveillance blood cultures [6,7,8]. However, it has been reported that catheter colonization plays a fundamental role in coagulase-negative staphylococci (CoNS) CRBSI [9,10,11,12]. A recent meta-analysis concludes that locking with an antimicrobial and low-dose heparin is more effective and safer when compared to other lock solutions (e.g., citrate, recombinant tissue plasminogen activator, ethylenediaminetetraacetic acid) to prevent CRBSI [13]. Some institutions have recommended antimicrobial lock (AL) prophylaxis in exceptional conditions [14,15]. Some evidence shows the benefit of this approach in reducing CRBSI, but patients are at risk of superinfection by resistant microorganisms and potential adverse events [4,8,16,17,18,19]. We conducted a prospective pilot study using surveillance by periodic extraction of quantitative blood culture (QBC) in an HD unit. The study aimed to prevent CRBSI by using pre-emptive antimicrobial locks in patients with TCVC-related colonization.

## 2. Materials and Methods

We conducted a prospective pilot study at the hemodialysis unit in the Clinica Universidad de Navarra, a 300-bed University Hospital in Pamplona, Spain. The hemodialysis unit serves an average of 44 patients, 22 patients per day with eleven dialysis machines, divided into two shifts for patient care. From March 2005 to May 2019, we included in the protocol all consecutive adult patients with a TCVC. Our institution performs surveillance quantitative blood cultures as part of the routine quality infection control program. We extracted ten milliliters of blood through each catheter lumen every 30 days. Blood was inoculated in lysis-centrifugation tubes (Isolator system; Wampole Laboratories, Cranbury, NJ, USA). We inverted gently five times to mix blood with the reactant. This procedure was intended to prevent coagulation and initiate red blood lysis. We transported the sample to the laboratory in order to process it without delay. We vortexed tubes for ten seconds and disinfected the stopper. Using a three milliliters syringe, we entered the upright tube at an angle so that the needle emerged from the bottom of the stopper between the wall of the tube and the side of the stopper. Then we tilted the tube to a horizontal position and collected the blood. We purged the air in the syringe and removed the needle. We inoculated 0.3 mL per plate (i.e., blood agar, chocolate agar, and Sabouraud dextrose agar) and incubated appropriately. We daily examined all plates. Plates were discarded after seven days of incubation if no growth was detected. We identified microorganisms isolated by VITEK-2 system or by matrix-assisted laser desorption ionization-time flight mass spectrometry (MALDI-TOF/MS) and performed broth microdilution susceptibility tests following the current European Committee on Antimicrobial Susceptible Testing (EUCAST) recommendations. All patients participated voluntarily and did not receive compensation. The participants provided informed consent. The study protocol was approved by the ethics committee of Navarra.

We designed an algorithm based on surveillance QBC to classify patients (Figure 1). We chose the cut-offs according to previous studies [11,20]. When surveillance QBC was negative (i.e., group one), we considered these patients at low risk of infection, and consequently they continued with the routine surveillance monitoring. Patients with CoNS counts < 100 CFU/mL (group two) were classified as having a moderate risk of infection (i.e., non-significant colonization) and they continued with the routine surveillance monitoring. We classified patients with CoNS counts ≥ 100 but <1000 CFU/mL (i.e., group three) as patients with a high risk of developing CRBSI (i.e., critical catheter colonization (CCC)) [6]. For CCC, we recommended performing pre-emptive antimicrobial locks. Given the duration of the study, the selection of the antibiotic was based on the antimicrobial susceptibility test and the best scientific evidence available at the time of recommendation. At the beginning of the study, Vancomycin was recommended, later we used Teicoplanin, and in recent years we have used Daptomycin. The nephrologist in charge of the patient could follow the recommendation (i.e., AL group) or could ignore it and keep patients with standard heparin locks (i.e., standard of care). Recovery of ≥1000 CFU/mL of a microorganism in a surveillance QBC (i.e., group four) was considered CRBSI [21,22] and treated according to guidelines [8]. Patients with isolation of *S. lugdunensis* or microorganisms other than CoNS were excluded from the present study; their management was individualized in each case.

We instilled five milliliters of lock solution at the end of every hemodialysis session. Locks were administered for 21 days (nine HD sessions).

Table 1 shows lock composition according to the antimicrobial used.

The study’s primary endpoint was to compare CRBSI incidence within 30 days after CCC diagnosis between AL and Standard of Care (SoC) groups. Relapse was defined as the isolation of any bacteria in QBC in the next three months of a CCC. All analyses were performed per protocol using SPSS software (version 26.0). We used non-parametric tests due to the sample size and the absence of normality. We compared the treatment groups’ CRBSI incidence and eradication ability with Fisher’s exact test; *p*-values < 0.05 were considered as statistically significant.

## 3. Results

### 3.1. Demographic

We included 149 patients in the surveillance program (Table 2). We excluded 29 patients because of isolation of *S. lugdunensis.* Detailed isolation of QBC is shown in Appendix A. The median follow-up for all patients was 8.80 months (IQR 2.82–26.72). Patients in group three and four were followed for 38.28 months (IQR 26.2–99) and 50.41 months (IQR 12.8–82.8), respectively. A total of 71 patients (47.7%) died during follow-up. No death was the result of a catheter-related infection episode.

### 3.2. Isolates

We collected a total of 3029 QBC with a median number per patient of nine (IQR, 3.5–26). Of these, 2762 (91.19%) were negative (i.e., group one), 159 (5.25%) had a non-significant colonization (i.e., group two), 86 (2.84%) had a CCC (i.e., group three), and 22 (0.73%) were diagnosed with CRBSI (all identified as *S. epidermidis*).

In group three, 66 isolates were identified as S. epidermidis, 11 as *S. haemolyticus*, 5 as *S. warneri*, 3 as *S. hominis*, and 1 as *S. capitis*. Regardless of the isolated microorganism, 45 (52.33%) isolates were treated with AL, and the remaining 41 (47.67%) were managed with SoC. The median time of AL duration was 19 days (IQR, 18–21).

### 3.3. Progression to Bacteremia

None of the 114 patients (2921 isolates) with negative or non-significant cultures (groups one and two) progressed to CRBSI during follow-up. Three patients with non-relevant colonization (<100 CFU/mL, group two) subsequently presented a CCC with the same microorganism.

None of the 45 CCC treated with AL progressed to bacteremia (Figure 2), but four out of the 41 CCC treated with the SoC progressed to catheter-related bloodstream infection (all with the same microorganism). This corresponds to a number needed to treat (NNT) of ten patients (95% CI, 5.26–100, *p* = 0.046) to avoid one episode of catheter-related bloodstream infection.

### 3.4. Eradication Ability

Regarding eradication ability, 84.5% of antimicrobial-locked catheters were relapse-free and remained QBC sterile for three months, while just 18.2% of the SoC group patients remained negative during the same time. AL of patients’ catheters with CCC is a protective factor against relapse of colonization or bacteremia for up to three months (RR 0.19, 95% CI, 0.11–0.33, *p* < 0.001). Patients reported no adverse events related to AL. In subsequent QBCs extracted, we did not identify isolates resistant to the antimicrobials used during the period studied.

## 4. Discussion

CRBSI is a critical problem in HD patients causing prolonged hospitalization, increased morbidity, mortality, and medical costs. Catheter removal is needed in some situations. However, removing these catheters can be difficult in hemodialysis patients with limited venous access. Around 30–40% [1] of patients starting hemodialysis with a TCVC rely on it as their lifesaving vascular access. In this study, we demonstrate that identifying patients with a high-risk colonization and using pre-emptive AL may reduce CRBSI rates.

We calculated an NNT to prevent a CRBSI episode of ten; this means that AL should be given to ten patients with CCC to avoid one CRBSI episode. In a meta-analysis, authors found that vancomycin as generalized prophylaxis for TCVT colonization in oncological patients had a reduction in CRBSI similar to our findings [23]. Arechabala et al. also found a reduction in the incidence of CRBSI between 60–70% when using prophylactic AL [4]. Another recent meta-analysis found a 32% reduction in the risk of CRBSI in patients treated with AL compared with those who received heparin locks [24]. In fact, given the high efficacy that lock therapy has demonstrated, the International Society for Infectious Diseases [14] recommended its prophylactic use when the prevalence of catheter-associated infection is high. Nevertheless, overuse of AL is a cause for concern from the cost perspective and because of the emergence of multidrug-resistant bacteria and adverse events [3,4,25]. The progression to bacteremia risk assessment is one of the main strengths of our study because it allows a rational use of antimicrobials with a high success rate. The fact that none of the patients treated with AL progressed to bacteremia, and the ability of the therapy to avoid recurrence of infection for at least three months of follow-up, demonstrates that pre-emptive therapy is probably superior to targeted therapy because acting on immature biofilms achieves a better sterilization [26]. The meta-analysis by Dang et al. [27] reported minimal or no significant differences in the colonization of catheters treated with AL or with heparin. Likewise, Campos et al. [28] were unable to sterilize CoNS biofilms with minocycline, as was the case also with Luther et al. [29] with vancomycin or linezolid. These results may differ from ours because lock therapy was attempted on mature biofilms. The longer the biofilm time, the higher the cell density, the more bacteria will be in a dormant phase, and the higher the production of extracellular polymeric substances. Extracellular polymeric substances allow the accumulation of enzymes capable of degrading antimicrobial in the extracellular matrix. All the above allow the development of *quorum sensing* capable of modulating the expression of virulence factors according to fluctuations in cell population density [30]. These are the main reasons why AL action on an immature biofilm is preferable.

Other authors [3,12] have already warned of the potential problems with using catheter monitoring to identify patients at risk of developing CRBSI. The criteria we used showed an excellent ability to differentiate between colonization episodes with risk of progressing to CRBSI and those episodes lacking risk. In our study, no low-risk colonization episode progressed to bacteremia. We found no other studies that have proven the harmlessness of mild colonization. Through risk stratification, a more rational use of resources can be made by saving antimicrobial doses, avoiding adverse effects, and decreasing the concern about the emergence of multidrug-resistant bacteria.

Also noteworthy is the sterilizing capacity of AL in appropriate patients that we demonstrate in our study. In eight out of ten colonization episodes with a risk of progression to CRBSI, we were able to prevent a relapse at least for three months. This could favor longer catheter lifespan and preservation of vascular access in complex patients. The overall lifespan of TCVC in our cohort is longer than that in the reports by other authors [31], probably because maintaining catheter sterility retains adequate flow rates and ensures optimal performance.

Nevertheless, our study has several limitations. It is a single-center, non-randomized study. In addition, the definitions of CCC and CRBSI used are easier to achieve than those reported by other authors [4,32]. Although our criteria increase the sensitivity of CRBSI detection, the significant reported differences are more difficult to find, which increases our findings’ strength but could complicate the comparison of our results. Regarding the eradication capacity, relapses were not considered according to the strain, so the reader should interpret the results cautiously. Further studies are required to validate our findings.

Additionally, it is possible that our better results may be due to detection bias, promoting increased eradication because the AL acts on immature biofilm, which is easier to remove [26,30].

## 5. Conclusions

Our study demonstrates that the prevention of CRBSI with CoNS is feasible by screening for high-risk colonization patients and using pre-emptive antimicrobial lock.

## Figures and Tables

**Figure 1 antibiotics-11-01692-f001:**
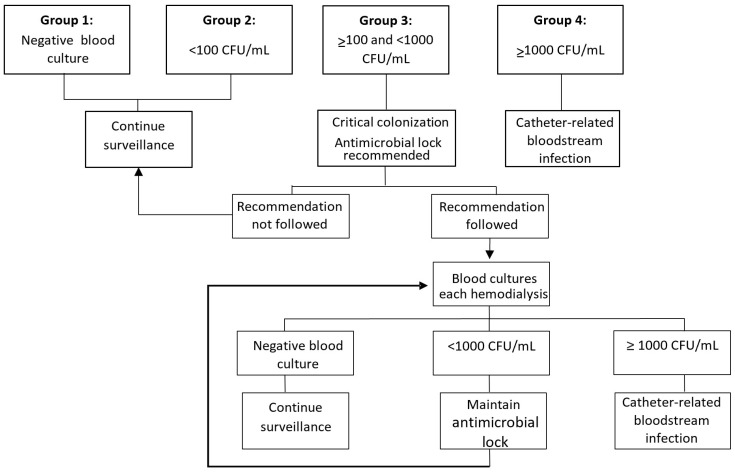
Algorithm for management of quantitative blood cultures results. Abbreviations: CFU, colony-forming unit; mL, milliliters.

**Figure 2 antibiotics-11-01692-f002:**
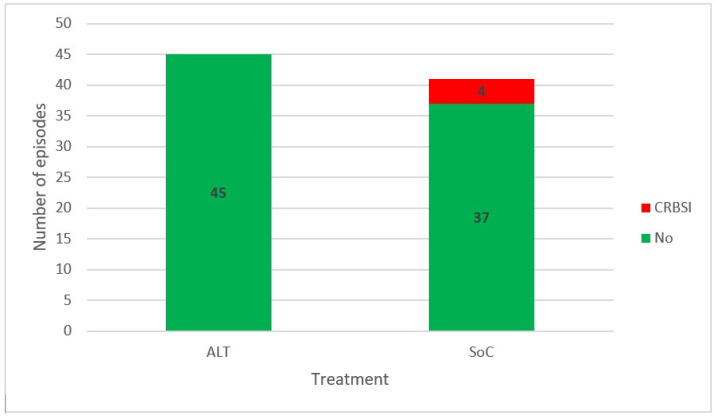
Progression to catheter-related bloodstream infection in the following 30 days of diagnosis. Abbreviations: ALT, antimicrobial lock therapy; SoC, standard of care; CRBSI, catheter-related bloodstream infection.

**Table 1 antibiotics-11-01692-t001:** Composition of the lock solutions used in the study.

Antimicrobial	Concentration (mg/mL)	Sodium Heparin Concentration (UI)
Teicoplanin	10	500
Daptomycin	5	500
Vancomycin	10	100
Standard of care	-	5000

**Table 2 antibiotics-11-01692-t002:** Characteristics of patients by quantitative blood culture result.

Variable	All Patients (n = 149)	At Least One CRBSI Patients (n = 22)	CCC Episode Patients (n = 13)	Never CCC or CRBSI Patients (n = 114)	*p*
Age in years, median (IQR)	68 (54–74.5)	69.5 (60.5–73.5)	72 (50–79.5)	67 (52.75–74)	0.437
Gender (male), n. (%)	85 (57)	8 (36.4)	5 (38.5)	72 (63.2)	**0.025**
Charlson score, median (IQR)	6 (5–8)	6.5 (5–8)	7 (5–8)	6 (5–8)	0.892
Time from TCVC insertion to beginning the study in months, median (IQR)	1.02 (0.5–1.88)	1.64 (1.26–16.04)	1.36 (0.78–3.19)	0.89 (0.46–1.52)	**<0.001**
Time from TCVC insertion to first CRBSI/CCC in months, median (IQR)	-	12.19 (5.22–27.68)	13.52 (4.44–62.83)	-	0.375
Lifespan of TCVC from insertion to removal in months, median (IQR)	7.72 (3.2–18.78)	31.02 (12.36–60.58)	9.89 (4.99–87.2)	5.98 (2.61–14.49)	**<0.001**
Follow-up in months, median (IQR)	8.80 (2.82–26.72)	50.41 (12.82–82.79)	38.28 (26.17–99)	4.94 (1.93–16.59)	**<0.001**
Deaths during follow-up, n. (%)	71 (47.7)	18 (81.8)	7 (53.8)	46 (40.4)	**0.001**
Time from CRBSI/CCC to death in months, median (IQR)	-	29.31 (11.06–61.93)	12.62 (2.76–42.12)	-	0.216
Ratio TCVC-patients	1.47	2.59	2	1.19	0.368
Cause of ESRD, n. (%)					0.533
-Glomerulonephritis	32 (21.47)	5 (22.72)	2 (15.4)	25 (21.9)	
-Diabetic kidney disease	20 (13.42)	3 (13.64)	4 (30.7)	13 (11.4)	
-Nephroangiosclerosis	19 (12.75)	2 (9.1)	1 (7.7)	16 (14)	
-Polycystic nephropathy	15 (10.07)	5 (22.72)	3 (23.1)	7 (6.1)	
-Others or unknown	63 (42.28)	7 (31.82)	3 (23.1)	53 (46.5)	

Abbreviations: CRBSI, catheter-related bloodstream infection; CCC, catheter critical colonization; IQR, interquartile range; n, number; TCVC, tunneled central venous catheter; ESRD, end-stage renal disease.

## Data Availability

The data that support the findings of this study are available at 10.6084/m9.figshare.21547893 (accessed on 21 November 2022).

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
