# Peer review of "Pre-Emptive Antimicrobial Locks Decrease Long-Term Catheter-Related Bloodstream Infections in Hemodialysis Patients"

_antibiotics, 2022, doi:10.3390/antibiotics11121692_

Round 1

Reviewer 1 Report

General comments

=============

Thank you for letting me peer-review your work! This paper is an interesting study for antimicrobial locks for long-term catheter-related bloodstream infections in hemodialysis patients.

Specific comments

=============

Major comments

---------------------

1. First of all, please change the position of the method section before the result section and after the introduction section.

2. In method section, please clarify the following points:

- Please add the detail of the study protocol approved by the institutional review board.

- Please clarify how this research got informed consent from each patient or waived.

- Please clarify the method of quantitative blood cultures.

- Please define CRBSI in this study.

- Please clarify how to choose antimicrobial (teicoplanin, daptomycin, vancomycin)

3. In result section

Please add the number of patients excluded (isolation of S.lugdunensis or other microorganisms). 

Minor comments

---------------------

4. Please modify the figure legend in Figure 1. It included medical abbreviation.

5. If available, please add the cause of death. Why did about half of the participants die during follow-up, except for CRBSI?

Reviewer 2 Report

This manuscript by Andres Blacon-Di Matteo et al. describes an interesting study on the value of locks in hemodialysis patients.

This manuscript deserves to be reviewed from a methodological/statistical point of view in order to obtain robust results.

Global: 

Prefer passive turns of phrase.

Numbers less than twelve should be written in full.

Italics: "i.e.", "e.g.", bacteria names, "et al.", "quorum sensing".

Specific

- Authors should be able to provide information on non-staphylococcal coagulase negative colonization/infection.

- How did the authors consider relapse/reinfection in their cohort?

- How did the authors account for multiple statistical test corrections?

- Can the authors stratify the results according to the antibiotic molecule used?

- Clarification is needed on antibiotic susceptibility testing, if appropriate antibiotic therapy was available, was an alternative available?

- Authors should justify the inclusion period/number of subjects to be included.

- The authors should provide details on the "technical standards" of isolation/identification.

- It is surprising that the authors chose to use a non-parametric test given the number of subjects included. Did they statistically assess the (non-)normality of the distribution?

- The data must be published, anonymized to comply with the law, but disclosure on request is not a valid reason.

Author Response

Thank you for your suggestions, please see the attachment. 

Round 2

Reviewer 1 Report

General comments

=============

Thank you for letting me peer-review your work! This paper is an interesting study for antimicrobial locks for long-term catheter-related bloodstream infections in hemodialysis patients.

Almost all responses were reasonable except for the following minor comment.

Minor comments

---------------------

In figures, medical abbreviations (QBC, ALT, SoC, CRBSI) should be explained in legends. 

Author Response

Dear Reviewer 1,

Thank you for the opportunity to make these corrections. We have fixed the issue.

Sincerely, 

Jose Luis Del Pozo

Reviewer 2 Report

The manuscript could be optimized answering these comments : 

- Authors should be able to provide information on non-staphylococcal coagulase negative colonization/infection. We appreciate the suggestion. We have excluded isolates of S. lugdunensis and other bacteria different from CoNS because we consider that their management should be different (i.e., these bacteria are probably not candidates for conservative treatment). However, below we attach a table with all isolates. If you think it may interest your journal's readers, we will be happy to add it to the supplementary material.

--> please perform.

- How did the authors consider relapse/reinfection in their cohort? We defined relapse as the isolation of any bacteria in the quantitative blood culture for three months after a catheter critical colonization episode. We have added definitions to the manuscript (Lines 114 – 115)

--> How have the authors taken into account the possible bias due to the reintroduction of particular strains. I suggest to suppress these patients from the analysis.

- Can the authors stratify the results according to the antibiotic molecule used? This is an exciting question that we will try to answer in a post-hoc study. Since it exceeds our research objectives, it is not included in the results. Thank you for being so interested.

--> I suggest to perform the analysis on the present data. Please perform.

- Authors should justify the inclusion period/number of subjects to be included. This is a pilot study. An adequate sample size calculation could not be made because there are no studies like ours.

--> Justify the period of inclusion.

--> Indicate the obtained statistical power with such a cohort.
